# Stochastic Nonparametric Event-Tensor Decomposition

**Shandian Zhe, Yishuai Du**
School of Computing, University of Utah
`zhe@cs.utah.edu, yishuai.du@utah.edu`

## Abstract

Tensor decompositions are fundamental tools for multiway data analysis. Existing approaches, however, ignore the valuable temporal information along with data, or simply discretize them into time steps so that important temporal patterns are easily missed. Moreover, most methods are limited to multilinear decomposition forms, and hence are unable to capture intricate, nonlinear relationships in data. To address these issues, we formulate event-tensors, to preserve the complete temporal information for multiway data, and propose a novel Bayesian nonparametric decomposition model. Our model can (1) fully exploit the time stamps to capture the critical, causal/triggering effects between the interaction events, (2) flexibly estimate the complex relationships between the entities in tensor modes, and (3) uncover hidden structures from their temporal interactions. For scalable inference, we develop a doubly stochastic variational Expectation-Maximization algorithm to conduct an online decomposition. Evaluations on both synthetic and real-world datasets show that our model not only improves upon the predictive performance of existing methods, but also discovers interesting clusters underlying the data.

## 1 Introduction

Tensors represent the high-order interactions between entities in multiway data. Such interactions are ubiquitous in real-world applications. For instance, in online shopping, *users* purchase *commodities* under different *web contexts* — these interactions can be represented by a three mode tensor *(user, commodity, web context)*. To analyze tensor data, we use decomposition approaches — where we jointly estimate a set of latent factors for each entity, and the mapping between the latent factors and tensor entry values. The latent factors can reveal hidden structures of the entities, such as clusters/communities; the mapping characterizes the entities' relationships (in terms of their factor representations), and can be used to predict missing entry values.

Despite the wide success of existing tensor decomposition algorithms (Tucker, 1966; Harshman, 1970; Kang et al., 2012; Choi and Vishwanathan, 2014), most methods assume a simple multilinear decomposition form, which might be insufficient to estimate intricate, nonlinear relationships in data. More important, most methods ignore the valuable temporal information along with data or exploit them in a relatively coarse way. For instance, the time stamp of each interaction is usually abandoned and only their counts are used for count tensor decomposition (Chi and Kolda, 2012; Hansen et al., 2015; Hu et al., 2015b). More elegant approaches (Xiong et al., 2010; Schein et al., 2015, 2016) discretize the time stamps into steps, *e.g.,* weeks/months, and use a set of time factors to represent each step. The tensor is hence augmented with a time mode. The decomposition may further use Markov assumptions to encourage smooth transitions between the time factors (Xiong et al., 2010). However, in each time step, the occurrences of the interactions are treated independently. Hence, important temporal patterns, such as causal/triggering effects in adjacent interactions, cannot be well modeled or captured.

To address these issues, we first formulate a new data abstraction, event-tensor, to preserve all the time stamps for multiway data. In an event-tensor, each entry comprises a sequence of interaction events rather than a numerical value. Second, we propose a powerful Bayesian nonparametric model to decompose event-tensors (Section 3). We hybridize latent Gaussian processes and Hawkes processes to capture various excitation effects among the observed interaction events, and the underlying complex relationships between the entities that participated in the events. Furthermore, we design a novel triggering function that enables discovering clusters of entities (or latent factors) in terms of excitation strengths. Besides, the triggering function allows us to flexibly specify the triggering range (say, via domain knowledge) to better capture local excitations and to control the trade-off to the computational cost. Finally, to handle data where both the tensor entries and interaction events are many, we derive a fully decomposed variational model evidence lower bound by using Poisson process superposition theorem and the variational sparse Gaussian process framework (Titsias, 2009). Based on the bound, we develop a doubly stochastic variational Expectation-Maximization algorithm to fulfill a scalable, online decomposition (Section 4).

For evaluation, we examined our model in both predictive performance and structure discovery. On three real-world datasets, our model often largely improves upon the prediction accuracy of the existing methods that use Poisson processes and/or time factors to incorporate temporal information. Simulation shows the latent factors estimated by our model clearly reflect the ground-truth clusters while by the competing methods do not. We further examined the structures discovered by our model on the real-world datasets and found many interesting patterns, such as groups of 911 accidents with strong associations, locations of townships that are apt to have consecutive accidents, and UFO shapes that are more likely to be sighted together (Section 6).

## 2 Background

**Tensor Decomposition.** We denote a $K$-mode tensor by $\mathcal{M} \in \mathbb{R}^{d_1 \times \cdots \times d_K}$, where $d_k$ is the dimension of $k$-th mode, corresponding to $d_k$ entities (*e.g.,* users or items). The entry value at location $\mathbf{i} = (i_1, \ldots, i_K)$ is denoted by $m_\mathbf{i}$. Given a tensor $\mathcal{W} \in \mathbb{R}^{r_1 \times \cdots \times r_K}$, and a matrix $\mathbf{U} \in \mathbb{R}^{s \times t}$, we can multiply $\mathcal{W}$ by $\mathbf{U}$ at mode $k$ when $r_k = t$. The result is a new tensor of size $r_1 \times \ldots \times r_{k-1} \times s \times r_{k+1} \times \ldots \times r_K$. Each entry is computed by $(\mathcal{W} \times_k \mathbf{U})_{i_1 \ldots i_{k-1} j i_{k+1} \ldots i_K} = \sum_{i_k=1}^{r_k} w_{i_1 \ldots i_K} u_{j i_k}$. For decomposition, we introduce $K$ latent factor matrices, $\mathcal{U} = \{\mathbf{U}^{(1)}, \ldots, \mathbf{U}^{(K)}\}$, to represent the entities in each tensor mode — each $\mathbf{U}^{(k)}(j, :)$ are the latent factors of the $j$-th entity in mode $k$.

The classical Tucker decomposition (Tucker, 1966) incorporates a small core tensor $\mathcal{W} \in \mathbb{R}^{r_1 \times \cdots \times r_K}$, and assumes $\mathcal{M} = \mathcal{W} \times_1 \mathbf{U}^{(1)} \times_2 \ldots \times_K \mathbf{U}^{(K)}$. We can simplify Tucker decomposition, by restricting $r_1 = \ldots = r_K$ and $\mathcal{W}$ to be diagonal. Then we reduce to CANDECOMP/PARAFAC (CP) decomposition (Harshman, 1970). While many other decomposition methods have been proposed *e.g.,* (Chu and Ghahramani, 2009; Kang et al., 2012; Choi and Vishwanathan, 2014), most of them are still based on the Tucker/CP forms. However, the multilinear assumptions might be insufficient to capture intricate, highly nonlinear relationships in data.

Recently, several Bayesian nonparametric tensor decomposition models (Xu et al., 2012; Zhe et al., 2016b) are proposed, which are flexible to capture various nonlinear relationships in data. For example, Zhe et al. (2016b) considered each entry value $m_\mathbf{i}$ as a function of the corresponding latent factors, *i.e.,* $m_\mathbf{i} = f([\mathbf{U}^{(1)}(i_1, :), \ldots, \mathbf{U}^{(K)}(i_K, :)])$, and placed a Gaussian process (GP) (Rasmussen and Williams, 2006) prior over $f(\cdot)$, to automatically infer the (possible) nonlinearity of $f(\cdot)$. These methods often improve the CP/Tucker decompositions by a large margin in missing value prediction.

**Decomposition with Temporal Information.** Practical tensors often come with temporal information, namely the time stamps of those interactions. For example, from a file access log, we can extract not only a three-mode *(user, action, file)* tensor, but also the time stamps for each user taking the action to access a file. To use the temporal information in the decomposition, many methods discard the time stamps, use a Poisson (process) likelihood to model the interaction frequency $m_\mathbf{i}$ in each entry $\mathbf{i}$, $p(m_\mathbf{i}) \propto e^{-\lambda_\mathbf{i} T} \lambda_\mathbf{i}^{m_\mathbf{i}}$ (Chi and Kolda, 2012; Hu et al., 2015b), and perform the Tucker/CP decomposition over $\{\lambda_\mathbf{i}\}$ or $\{\log(\lambda_\mathbf{i})\}$. More refined approaches (Xiong et al., 2010; Schein et al., 2015, 2016) first discretize the time stamps into several steps, such as months/weeks, and augment the original tensor with a time mode. Then a time factor matrix $\mathbf{T}$ are estimated in the decomposition. While the interactions from different time steps are modeled with distinct time factors, the ones in the same interval are considered independently (given the latent factors), say, being modeled by Poisson likelihoods (Schein et al., 2015, 2016). A Markov assumption might be used to encourage

the smoothness between the time factors. For example, Xiong et al. (2010) assigned a conditional Gaussian prior over each $\mathbf{T}(k,:)$, $p\big(\mathbf{T}(k,:)|\mathbf{T}(k-1,:)\big) = \mathcal{N}\big(\mathbf{T}(k,:)|\mathbf{T}(k-1,:),\sigma^2\mathbf{I}\big)$.

## 3 Model

Despite the success of existing approaches in exploiting temporal information, they entirely drop the time stamps and hence are unable to capture the important, triggering or causal effects between the interactions. The triggering effects are common in real-world applications. For example, the event that *user A* purchased *commodity B* may excite *A*'s friend *C* to purchase *B* as well. The triggering effects are usually local and decay fast with time; dropping the time stamps and considering the event occurrences independently make us unable to model/capture these effects.

To address these issues, and hence to further capture the complex relationships and important structures underlying the interaction events, we formulate a new data abstraction, event-tensor, to preserve all the time stamps. We then propose a powerful Bayesian nonparametric model to decompose the event-tensors, discussed as follows.

### 3.1 Event-Tensor Formulation

First, let us look at the definition of event-tensors. To preserve the complete temporal information in decomposition, we relax the definition that tensors must be multidimensional arrays of numerical values. Instead, we define that each entry is a sequence of events, *i.e.,* $m_\mathbf{i} = \{s_\mathbf{i}^1,\ldots,s_\mathbf{i}^{n_\mathbf{i}}\}$ where each $s_\mathbf{i}^k(1 \le k \le n_\mathbf{i})$ is a time stamp when the interaction happened, and $n_\mathbf{i}$ the count of the events. Note that, different entries correspond to distinct types of interaction events, since the involved entities (or latent factors) are different. We name this tensor as an event-tensor. Given the observed entries $\{m_\mathbf{i}\}$, we can flatten their event sequences to obtain a single sequence $S = [(s_1,\mathbf{i}_1),\ldots(s_N,\mathbf{i}_N)]$ where $s_1 \le \ldots \le s_N$ are all the time stamps, and each $\mathbf{i}_k$ is the entry index for the event $s_k (1 \le k \le N)$ .

### 3.2 Nonparametric Event-Tensor Decomposition

Now, we consider a probabilistic model for event-tensor decomposition. While Poisson processes (PPs) have many nice properties and are often good choices of modeling events (Schein et al., 2015), they assume event occurences are independent (*i.e.,* independent increments), and hence are unable to capture the influences of the events on each other. To overcome this limit, we use a much more expressive point process, Hawkes process (Hawkes, 1971), for events modeling in tensor entries. Given an event sequence $\{t_1,\ldots t_n\}$, the Hawkes process defines the event rate $\lambda$ as a function of time $t$, $\lambda(t) = \lambda_0 + \sum_{t_i < t} h(t - t_i)$, where $\lambda_0$ is the base rate (or background rate), and $h(\Delta_t)$ is the triggering function, which describes the strength of a preceding event triggering a new event at time $t$. Note that the strength usually decays with time. For example, a commonly used triggering function is $h(\Delta_t) = \beta\exp(-\frac{\Delta_t}{\tau})$, which expresses an exponential decay over time. The joint probability of the sequence $\{t_1,\ldots t_n\}$ is $p(\{t_1,\ldots t_n\}) = e^{-\int_0^T \lambda(t)}\prod_{j=1}^n \lambda(t_j)$, where $T$ is the total time span.

In our model, for each observed entry $\mathbf{i}$ we use a Hawkes process to sample the interaction sequence $m_\mathbf{i}$. As in section 3.1, we denote the flattened single event sequence over all the observed entries by $S = [(s_1,\mathbf{i}_1),\ldots,(s_N,\mathbf{i}_N)]$. For the process in entry $\mathbf{i}$, we define the rate function as

$$\lambda_\mathbf{i}(t) = \lambda_\mathbf{i}^0 + \sum_{s_n < t} h_{\mathbf{i}_n \to \mathbf{i}}(t - s_n) \tag{1}$$

where $\lambda_\mathbf{i}^0$ is the base rate and $h_{\mathbf{i}_n \to \mathbf{i}}(\Delta_t)$ is the triggering function.

Now, let us present the detailed design for the base rate and triggering function. First, to capture the (complex) relationships between the entities underlying the events in entry $\mathbf{i}$, we assume the background rate $\lambda_\mathbf{i}^0$, is a (possible) nonlinear function of the corresponding latent factors, $\mathbf{x}_\mathbf{i} = [\mathbf{U}^{(1)}(i_1,:),\ldots,\mathbf{U}^{(K)}(i_K,:)]$. To ensure the positiveness of $\lambda_\mathbf{i}^0$, we sample a latent function $f(\mathbf{x}_\mathbf{i})$ and take $\lambda_\mathbf{i}^0 = e^{f(\mathbf{x}_\mathbf{i})}$. We place a Gaussian process (GP) prior over $f(\cdot)$. Hence, the latent function values $\mathbf{f}$ for all the observed entries follow a multivariate Gaussian distribution,

$$p(\mathbf{f}|\mathcal{U}) = \mathcal{N}\big(\mathbf{f}|\mathbf{0}, c(\mathbf{X},\mathbf{X})\big) \tag{2}$$

where each row of the input matrix $\mathbf{X}$ corresponds to one entry, and are the concatenation of the corresponding latent factors; $c(\cdot,\cdot)$ is the covariance function, and can be some nonlinear or/and periodical kernels.

Second, to capture various excitation effects and the underlying structures of the entities, we design the triggering function as the following form:

$$h_{\mathbf{i}_n \to \mathbf{i}}(t - s_n) = k(\mathbf{x}_{\mathbf{i}_n}, \mathbf{x}_{\mathbf{i}}) h_0(t - s_n) \tag{3}$$

where $k(\cdot, \cdot)$ is a kernel function, $\mathbf{x}_{\mathbf{i}_n}$ and $\mathbf{x}_{\mathbf{i}}$ are the concatenated latent factors for entries $\mathbf{i}_n$ and $\mathbf{i}$, respectively; $h_0(\cdot)$ is the base triggering function which we will explain later. In our design, the excitation strength between the two types of interactions, $\mathbf{i}_n$ and $\mathbf{i}$, is determined by the closeness/similarity between the associated entities. The closeness is measured by the kernel function of their latent factors. Such design enables our model to discover the grouping structures hidden in the triggering effects — entities in the same group/community more strongly excite each other to interact with other modes' entities from the same group, *e.g.,* "purchasing the same brand of products" and "watching the same types of movies".

Next, we design a local base triggering function, to better capture the locality of the triggering effects,

$$h_0(t - s_n) = \mathbb{1}(s_n \in A_t) \beta e^{-\frac{1}{\tau}(t - s_n)} \tag{4}$$

where $A_t$ is the set of possible triggering events to time $t$. By setting $A_t$, we can specify the appropriate range of triggering effects through domain knowledge, or the best trade-off to the computational efficiency. In our model, we define $A_t$ as the collection of preceding events nearest to time $t$ in the time window $\Delta_{\max}$, $A_t = \{s_j | s_j \in P_t(C_{\max}), t - \Delta_{\max} \le s_j \le t\}$ where $P_t(C_{\max})$ are $C_{\max}$ preceding events nearest to $t$.

Finally, given the observed entries $\{m_{\mathbf{i}}\}$, based on (1) and (2), the joint probability of our model is

$$p(\{m_{\mathbf{i}}, f_{\mathbf{i}}\} | \mathcal{U}) = \mathcal{N}(\mathbf{f} | \mathbf{0}, c(\mathbf{X}, \mathbf{X})) \prod_{\mathbf{i}} e^{-\int_0^T \lambda_{\mathbf{i}}(t) \mathrm{d}t} \prod_{j=1}^{n_{\mathbf{i}}} \lambda_{\mathbf{i}}(s_{\mathbf{i}}^n) \tag{5}$$

where $m_{\mathbf{i}} = \{s_{\mathbf{i}}^1, \ldots, s_{\mathbf{i}}^{n_{\mathbf{i}}}\}$ and $f_{\mathbf{i}}$ is the latent function value for entry $\mathbf{i}$, used in our definition of the base rate, $\lambda_{\mathbf{i}}^0 = e^{f_{\mathbf{i}}}$.

# 4 Algorithm

## 4.1 Decomposable Variational Lower Bound

Exact inference of our model is computationally infeasible for large data, because the GP term in (5) is required to compute the covariance matrix $c(\mathbf{X}, \mathbf{X})$ and its inverse, which intertwine all the latent factors — when the number of observed entries is large, the computation is infeasible. Furthermore, the log joint probability of our model involves many log-summation terms, $\{\log(\lambda_{\mathbf{i}}(s_{\mathbf{i}}^n))\}$ — these terms further couples the latent factors (see (3)) and the base triggering function parameters, $\beta$ and $\tau$ (see (4)), making the computation even less efficient.

To tackle these problems, we first consider the standard variational sparse GP framework (Titsias, 2009). We introduce $Q$ pseudo inputs $\mathbf{B}$ and targets $\mathbf{g}$. Note that $Q$ is much smaller than the number of tensor entries. We assume $\mathbf{g}$ and $\mathbf{f}$ are sampled from the same Gaussian process, and hence they jointly follow a multivariate Gaussian distribution, $p(\mathbf{f}, \mathbf{g}) = \mathcal{N}([\mathbf{g}; \mathbf{f}] | \mathbf{0}, \mathbf{C})$ where $\mathbf{C} = [c(\mathbf{B}, \mathbf{B}), c(\mathbf{B}, \mathbf{X}); c(\mathbf{X}, \mathbf{B}), c(\mathbf{X}, \mathbf{X})]$ and $c(\mathbf{X}, \mathbf{B})$ is the cross-covariance between $\mathbf{X}$ and $\mathbf{B}$. We then augment our model with the pseudo target $\mathbf{g}$, $p(\mathbf{f}, \mathbf{g}, \{m_{\mathbf{i}}\} | \mathcal{U}) = p(\mathbf{g}) p(\mathbf{f} | \mathbf{g}) p(\{m_{\mathbf{i}}\} | \mathbf{f}, \mathcal{U})$. Following (Titsias, 2009), we introduce a variational distribution $q(\mathbf{g})$ and apply Jensen's inequity to obtain $\log(p(\{m_{\mathbf{i}}\} | \mathcal{U})) \ge \mathbb{E}_{q(\mathbf{g})} \log(\frac{p(\mathbf{g})}{q(\mathbf{g})}) + \mathbb{E}_{q(\mathbf{g})} \log(\mathbb{E}_{p(f|g)} p(\{m_{\mathbf{i}}\} | \mathbf{f}, \mathcal{U}))$. Next, we apply Jensen's inequality again for the second term to switch the order of logarithm and the expectation, so as to obtain a lower bound decomposed over tensor entries, $\log(\mathbb{E}_{p(f|g)} p(\{m_{\mathbf{i}}\} | \mathbf{f}, \mathcal{U})) \ge \mathbb{E}_{p(f|g)} \log(p(\{m_{\mathbf{i}}\} | \mathbf{f}, \mathcal{U})) = \sum_{\mathbf{i}} \mathbb{E}_{p(f_{\mathbf{i}} | g)} \log(p(m_{\mathbf{i}} | f_{\mathbf{i}}, \mathcal{U}))$. Note that $p(f_{\mathbf{i}} | g)$ is scalar conditional Gaussian distribution. However, this step is infeasible, because the expectations are not analytical — each base rate $e^{f_{\mathbf{i}}}$ is trapped in a set of the log-summation terms, *i.e.,* $\log(p(m_{\mathbf{i}} | f_{\mathbf{i}}, \mathcal{U})) = -e^{f_{\mathbf{i}}} T + \sum_{j=1}^{n_{\mathbf{i}}} \log(e^{f_{\mathbf{i}}} + a_j) + a_0$ where $a_0, \{a_j\}$ are the terms irrelevant to $f_{\mathbf{i}}$. This stems from the additive form of the Hawkes process rate function (see (1)). The expectation w.r.t a Gaussian distribution is not analytical.

To solve this problem, we exploit Poisson process super-position theorem (Cinlar and Agnew, 1968) to further augment our model with event cause variables. Thereby the base rate can be decoupled from the log-summation terms, and we can derive a tractable and decomposable bound.

Specifically, by the super-position theorem, each additive component in the rate function (1) can be considered as an independent Poisson process. The Hawkes process is equivalent to the union of

these Poisson processes — each event is sampled from one of these processes. Therefore, it is natural to introduce a latent cause variable $z$ for each event: $z = 0$ if the event is caused by the base rate; $z = n$ if the event is caused by a preceding event $s_n$.

For clearer description, let us consider the flattened single event sequence $S = [(s_1, \mathbf{i}_1), \ldots, (s_N, \mathbf{i}_N)]$ over all the observed tensor entries. Note that this is an equivalent representation of our observed data $\{m_{\mathbf{i}}\}$. For each event $s_j (1 \leq j \leq N)$, we introduce a latent cause variable $z_j$. Thanks to our local triggering function (4), we can use domain knowledge to determine an appropriate range of the cause, *i.e.*, $z_j \in \{0\} \cup \bar{A}_{s_j}$ where $\bar{A}_{s_j}$ are the indices of the events in $A_{s_j}$. Note that short ranges are helpful to capture local excitations and to reduce the computation cost in model estimation. Given the latent cause variables, $\{z_j\}_{j=1}^N$, we can partition the whole sequence $S$ into multiple Poisson process sequences. The probability of our model augmented with the latent cause variables is then derived by

$$p(\{s_j, \mathbf{i}_j, z_j\}_{j=1}^N | \mathbf{f}) = \prod_{\mathbf{i} \in \mathcal{O}} p(\{s_n | \mathbf{i}_n = \mathbf{i}, z_n = 0\} | \lambda_{\mathbf{i}}^0) \prod_{j=1}^N \prod_{\mathbf{i} \in \mathcal{O}} p(\{s_n | \mathbf{i}_n = \mathbf{i}, z_n = j\} | h_{\mathbf{i}_j \rightarrow \mathbf{i}}(t - s_j))$$

$$= \prod_{\mathbf{i} \in \mathcal{O}} \prod_{j=1}^N e^{-\int_{s_j}^T h_{\mathbf{i}_j \rightarrow \mathbf{i}}(t - s_j) \mathrm{d}t} \prod_{j=1}^N \prod_{n \in \bar{A}_{s_j}} h_{\mathbf{i}_n \rightarrow \mathbf{i}_j}(s_j - s_n)^{\mathbb{1}(z_j = n)} \prod_{\mathbf{i} \in \mathcal{O}} e^{-T \cdot e^{f_{\mathbf{i}}} + f_{\mathbf{i}} \left( \sum_{j=1}^N \mathbb{1}(\mathbf{i}_j = \mathbf{i}, z_j = 0) \right)}$$

where $\lambda_{\mathbf{i}}^0 = e^{f_{\mathbf{i}}}$, $\mathcal{O}$ are the indices for all the observed entries, and $\mathbb{1}(\cdot)$ is the indicator function. As we can see, the latent function values $\mathbf{f}$ are now decoupled into individual exponential terms. Although $e^{-T \cdot e^{f_{\mathbf{i}}}}$ looks nontrivial, we are still able to follow (Titsias, 2009) to derive an analytical variational bound (given in the below).

To infer the posterior distributions of the latent causes $\mathbf{z} = [z_1, \ldots z_N]$, we further introduce a variational posterior $q(\mathbf{z})$ with the mean-field form, $q(\mathbf{z}) = \prod_{j=1}^N q(z_j)$. Using the standard frameworks for variational sparse GP and mean-field approximations, we finally derive a tractable variational model evidence lower bound,

$$\mathcal{L} = -\sum_{j=1}^N \sum_{\mathbf{i} \in \mathcal{O}} \int_{s_j}^T h_{\mathbf{i}_j \rightarrow \mathbf{i}}(t - s_j) \mathrm{d}t + \sum_{j=1}^N \mathbb{E}_{q(\mathbf{g})} \mathbb{E}_{p(f_{\mathbf{i}_j} | \mathbf{g})} \left( \mathbb{E}_{q(z_j)} \left( \mathbb{1}(z_j = 0) \right) f_{\mathbf{i}_j} - \frac{T}{n_{\mathbf{i}_j}} e^{f_{\mathbf{i}_j}} \right)$$

$$+ \sum_{j=1}^N \sum_{n \in \bar{A}_{s_j}} \mathbb{E}_{q(z_j)} \left( \mathbb{1}(z_j = n) \right) \log \left( h_{\mathbf{i}_j \rightarrow \mathbf{i}}(s_j - s_n) \right) + \mathbb{E}_{q(\mathbf{g})} \left( \log \frac{p(\mathbf{g})}{q(\mathbf{g})} \right) \tag{6}$$

where $q(\mathbf{g})$ is the variational posterior of the pseudo targets in sparse GP. We assume $q(\mathbf{g}) = \mathcal{N}(\mathbf{g} | \mu, \Sigma)$, and each $q(z_j)$ is a multinomial distribution. As we can see, the variational lower bound is decomposed over each event — this additive structure enables us to develop a scalable stochastic inference algorithm, presented as follows.

### 4.2 Doubly Stochastic Variational Expectation-Maximization Inference

Given the variational lower bound (6), our model inference amounts to maximizing this bound. While the standard variational Expectation-Maximization (EM) algorithm is available, this batch inference paradigm can be very inefficient when the observed events are many, because each E-M iteration requires to pass all the events. Moreover, the batch inference is not suitable for dynamic event-tensors, where the events are collected in real-time. It is therefore natural to design a stochastic inference algorithm, where we sample a mini-batch of events at a time and perform a local variational EM update: in the E step, we update the variational posteriors for the latent causes variables, $\{q(z_j)\}$. In the M step, we update the latent factors and all the other parameters with stochastic gradient accent.

However, this stochastic inference following the standard recipe may still be inefficient when the tensor entries are many as well. The reason is that our bound has a double summation term, $\sum_{j=1}^N \sum_{\mathbf{i} \in \mathcal{O}} \int_{s_j}^T h_{\mathbf{i}_j \rightarrow \mathbf{i}}(t - s_j) \mathrm{d}t$. While each time we only need to handle a mini-batch of events, for each event we have to process all the tensor entries in the inner summation. Consider that a small tensor of size $100 \times 100 \times 100$ can have up to 1 million observed entries. The computation can be extremely expensive.

To deal with both large numbers of events and tensor entries, we further develop a doubly stochastic variational EM algorithm. Specifically, we randomly partition both the events and the tensor entries

into mini-batches, $\{N_k\}$ and $\{M_l\}$, according to which we arrange our variational bound as

$$\mathcal{L} = \mathbb{E}_{q(\mathbf{g})}\big(\log \frac{p(\mathbf{g})}{q(\mathbf{g})}\big) + \sum_k \frac{|N_k|}{N} \sum_{j \in N_k} \phi_{s_j, \bar{A}_{s_j}} \frac{N}{|N_k|} + \sum_k \sum_l \frac{|N_k|}{N} \frac{|M_l|}{M} \sum_{j \in N_k} \sum_{\mathbf{i} \in M_l} \psi_{s_j, \mathbf{i}, \mathbf{i}_j} \frac{N}{|N_k|} \frac{M}{|M_l|}$$

where $|\cdot|$ is the size of the mini-batch, $M$ is the number of observed entries, $\psi_{s_j, \mathbf{i}, \mathbf{i}_j} = -\int_{s_j}^{T} h_{\mathbf{i}_j \to \mathbf{i}}(t - s_j)\mathrm{d}t$, and $\phi_{s_j, \bar{A}_{s_j}} = \mathbb{E}_{q(\mathbf{g})}\mathbb{E}_{p(f_{\mathbf{i}_j}|\mathbf{g})}\big(\mathbb{E}_{q(z_j)}(\mathbb{1}(z_j = 0))f_{\mathbf{i}_j} - \frac{T}{n_{\mathbf{i}_j}}e^{f_{\mathbf{i}_j}}\big) + \sum_{n \in \bar{A}_{s_j}} \mathbb{E}_{q(z_j)}\big(\mathbb{1}(z_j = n)\big)\log\big(h_{\mathbf{i}_j \to \mathbf{i}}(s_j - s_n)\big)$. Then, the bound can be considered as an expectation of a stochastic objective, $\mathcal{L} = \mathbb{E}_{p(k), p(l)}(\tilde{\mathcal{L}}_{k,l})$, where $p(k) = \frac{|N_k|}{N}$, $p(l) = \frac{|M_l|}{M}$, and

$$\tilde{\mathcal{L}}_{k,l} = \mathbb{E}_{q(\mathbf{g})}\big(\log \frac{p(\mathbf{g})}{q(\mathbf{g})}\big) + \sum_{j \in N_k} \phi_{s_j, \bar{A}_{s_j}} \frac{N}{|N_k|} + \sum_{j \in N_k} \sum_{\mathbf{i} \in M_l} \psi_{s_j, \mathbf{i}, \mathbf{i}_j} \frac{N}{|N_k|} \frac{M}{|M_l|}.$$

We can therefore develop a doubly-stochastic EM algorithm to maximize $\mathcal{L}$. Each time, we sample two mini-batches, $N_k$ and $M_l$, one for the events and the other for the tensor entries. We then optimize the stochastic objective, $\tilde{\mathcal{L}}_{k,l}$, with one E-M iteration. In the E step, we optimize the variational posteriors of the latent causes $\{q(z_j)\}$ associated with the events in $N_k$; in the M step, we update all the other parameters $\boldsymbol{\theta}$ with stochastic gradient accent, $\boldsymbol{\theta} \leftarrow \boldsymbol{\theta} + \eta\frac{\partial \tilde{\mathcal{L}}_{k,l}}{\partial \boldsymbol{\theta}}$, where $\eta$ is the step size. Here $\boldsymbol{\theta}$ include the latent factors $\mathcal{U}$, the base triggering function parameters $\beta$ and $\tau$, the pseudo inputs $\mathbf{B}$, the kernel parameters, and the mean and covariance of $q(\mathbf{g})$. The detailed updating equations are listed in the supplementary material. Note that we cannot update $q(\mathbf{g})$ in the E-step because we do not have an analytical updating formula. We repeat this process until convergence or the maximum number of batches have been processed.

### 4.3 Algorithm Complexity

The time complexity of our algorithm is $O(Q^3 E_b + E_b V_b)$ where $E_b$ and $V_b$ are mini-batch sizes for events and tensor entries, respectively. Since $Q \ll N, M$ is constant, the time complexity is proportional to the sizes of the mini-batches. The space complexity is $O(\sum_{k=1}^{K} d_k r_k + Q^2)$, which is to store all the latent factors, and covariance of $q(\mathbf{g})$ and all the other parameters.

## 5 Related Works

Many excellent works have been proposed for tensor decomposition (Shashua and Hazan, 2005; Chu and Ghahramani, 2009; Sutskever et al., 2009; Acar et al., 2011; Hoff, 2011; Kang et al., 2012; Yang and Dunson, 2013; Rai et al., 2014; Choi and Vishwanathan, 2014; Hu et al., 2015a; Rai et al., 2015). Most of them are based on the classical, multilinear Tucker (Tucker, 1966) or CP (Harshman, 1970) decompositions. Recently, several nonparametric decomposition methods (Xu et al., 2012; Zhe et al., 2015, 2016a,b) were developed to capture nonlinear relationships in data, and have shown excellent predictive performance. However, most methods ignore the temporal information, or simply integrate them into count tensors (Chi and Kolda, 2012; Hansen et al., 2015; Hu et al., 2015b). The latter approaches usually use Poisson processes to model events, and ignore the temporal influences among those events. More elegant, temporal decomposition approaches (Xiong et al., 2010; Schein et al., 2015, 2016) introduce extra time factors to capture refined temporal patterns. However, since they discretize the time stamps into steps, they still lose information and are unable to capture fine-grained, triggering effects within the events. To address these problems, we formulated event-tensors to keep the complete temporal information, and proposed a powerful nonparametric event-tensor decomposition model by hybridizing latent GPs and Hawkes processes. Our model can be further extended for more general, temporal high-order relation data analysis (DuBois and Smyth, 2010; DuBois et al., 2013).

Due to the great flexibility, Hawkes processes (HPs) have been an important tool for discovering latent structures/relationships within general types of events, including reciprocal relationship on graphs (Blundell et al., 2012), latent network structures (Linderman and Adams, 2014), temporal clustering of documents (Du et al., 2015), network structures and topics in text-based cascades (He et al., 2015), user activity levels (Wang et al., 2017), etc. Moreover, many works have been developed for general HP modeling and inference (Zhou et al., 2013; Xu et al., 2016, 2017). Different from these methods, our doubly stochastic variational EM inference is designed for a hybrid of latent GP and HP model (on event-tensors).

## 6 Experiment

### 6.1 Predictive Performance

**Datasets**. To examine the predictive performance, we used three real-world datasets, *Article*(`www.kaggle.com/gspmoreira/articles-sharing-reading-from-cit-deskdrop/data`), *UFO*(`www.kaggle.com/NUFORC/ufo-sightings/data`) and *911*(`www.kaggle.com/mchirico/montcoalert/data`). The *Article* data are 12 month logs (03/2016 - 02/2017) of CI&T's Internal Communication platform (DeskDrop), which record users' operations on the shared articles, such as LIKE, FOLLOW and BOOKMARK. We extracted a three mode event-tensor *(user, operation, session id)*, of size $1895 \times 5 \times 2987$. There are $50,938$ entries observed to have events. The length of the longest event sequence in all the entries is 76. The total number of events is $72,312$. The *UFO* data consist of reported UFO sightings over the last century in the world, from which we extracted a two mode event-tensor *(UFO shape, city)*, of size $28 \times 19,408$, with $45,045$ entries observed to have sighting events. The longest event sequence length is 113. There are in total 77,747 events. The *911* data record the emergency (911) calls from 2015-12-10 to 2017-04-10 in Montgomery County, PA. We focused on the Emergence Medical Service (EMS) calls and extracted a two mode event-tensor *(EMS title, township)*, which is $72 \times 69$. There are $2,494$ entries observed to have events. The length of the longest event sequence is $545$. The total number of events are 59,270.

**Competing methods.** We compared our approach with the following typical methods to incorporate the temporal information into tensor decomposition. (1) CP-PTF — the Poisson process (PP) tensor factorization model using CP to decompose event rates. Similar to our approach, the CP form is applied in the log-domain to ensure the positiveness of the rates. We have investigated alternative methods (Chi and Kolda, 2012) where the latent factors are constrained to be nonnegative and so CP is directly applied over the rates. There was tiny difference in predictive performance. (2) CPT-PTF — where, similar to (Schein et al., 2015), we discretized the time stamps into multiple steps, augmented the tensor with a time mode to represent the time steps, and used PPs to model the event rate in the each step and CP to decompose the rates. As in (Xiong et al., 2010), we assigned conditional priors over the time factors to encourage their smooth transitions. In addition, we implemented (3) GP-PTF, the PP tensor factorization using GPs to model the event rates as a (nonlinear) function of the latent factors. This is the same strategy as we used to model the base rates of the HPs in our approach. For a fair comparison, we ran all the competing methods with standard stochastic inference, where each time, a mini-batch of tensor entries are sampled and the latent factors are updated with the stochastic gradient ascent. For GP-PTF, we used the same variational sparse GP framework as in our approach.

**Parameter settings.** We varied the number of latent factors from $\{1, 2, 5, 8\}$. For both GP-PTF and our method, we used the ARD kernel and set the number of pseudo inputs to 100. For a fair comparison, all the methods were initialized with the same latent factors, which were drawn element-wisely from the uniform distribution in $[0, 1]$. For training, we used the first $50K$, $40K$ and $40K$ events in *Article*, *UFO* and *911* respectively, and the remaining $22.3K$, $19.3K$ and $30.4K$ events for testing. For CPT-PTF, we varied the number of time steps from $\{5, 10, 20, 30\}$. For our approach, to examine different settings of the triggering range, we fixed the maximum number of triggering events $C_{\max}$ to 300 and varied the maximum triggering time window $\Delta_{\max}$ from $\{1, 2, 3\}$ hours for *Article* and *911*, and $\{1, 3, 5\}$ days for *UFO*. The mini-batch sizes of tensor entries (for all the methods), and events (for our method only) are both set to 100. We used AdaDelta (Zeiler, 2012) to adjust the step-size for the stochastic gradient ascent, and ran 100 epochs for each method. To remove the vibration of the prediction accuracy (due to the stochastic updates) from evaluation, we computed the test log likelihood after each epoch, and then reported the largest one as the prediction result.

**Results.** As shown in Fig. 1a-c, our approach outperforms all the competing methods, and in many cases improves them by a large margin. Note that the second best approach is always GP-PTF, implying complex, nonlinear relationships within the events. Furthermore, our improvement over GP-PTF demonstrates the advantage of using HPs to capture the (local) triggering effects between the events. To examine the dynamic behaviors of our doubly stochastic algorithm, we reported the test log likelihoods after each epoch in *Article* and *911* when the factor number was set to 8. As shown in Fig. 1d-e, the predictive performance of our algorithm kept improving and tended to converge at last. The running time is provided in the supplementary material.

### 6.2 Latent Structure Discovery

Next, we examined the capability of our model in discovering latent structures. We first simulated a small $10 \times 10 \times 10$ event-tensor with highly nonlinear hidden relationships between the latent factors, and the factors in each mode form 2 clusters (see the details in the supplementary

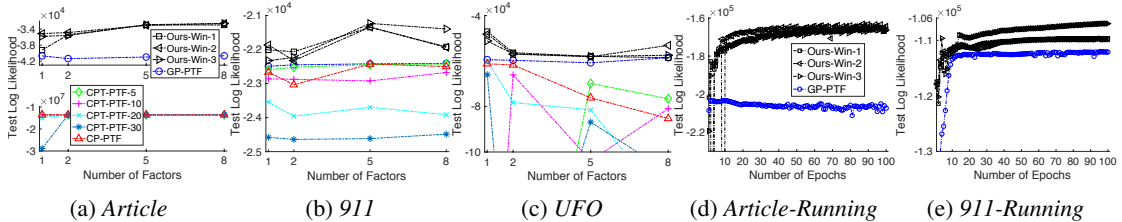

(a) *Article*     (b) *911*     (c) *UFO*     (d) *Article-Running*     (e) *911-Running*

Figure 1: The prediction performance on the three real-world datasets (a-c) and along with running time (d-e). CPT-PTF-$\{5, 10, 20, 30\}$ correspond to CPT-PTF using $\{5, 10, 20, 30\}$ time steps. Ours-Win-$\{1, 2, 3\}$ are our methods using three triggering time windows.

material). We ran CP-PTF, GP-PTF and our approach for 50 epochs to estimate the latent factors. The results of the second mode are reported in Figure 2. The markers (and the colors) of the points (*i.e.,* latent factors) exhibit their ground-truth classes. As we can see, CP-PTF obtained factors with mixed classes and unclear structures (Fig. 2a), GP-PTF with clearer cluster structures but mistaken groups (Fig. 2b). Our approach recovered both the clear cluster structures and correct factor groups (Fig. 2c).

In addition, we examined the structures discovered by our model from real-world applications. To this end, we used k-means plus BIC to cluster our estimated latent factors for *911* and *UFO* datasets (see Sec. 2.3 of the supplementary material for more details). We obtained 6 groups of EMS titles and 10 groups of townships for *911*, as shown in Figure 3a and c. We obtained 4 groups of UFO shapes, as shown in Figure 3b. Due to the space limit, we do not report the clusters of UFO sighting cities ($19K$ cities).

As we can see, the estimated latent factors for both *911* and *UFO* datasets reflect clear cluster structures, which may imply interesting patterns. First, we found that the clusters of EMS titles often contain accidents/events with strong associations. For example, Cluster 1 in Fig. 3a consist of {SHOOTING, AMPUTATION and S/B AT HELICOPTER LANDING} — after SHOOTING or accidental AMPU-TATION, the urgent rescue may require HELICOPTER supports. For another example, Cluster 2 are about disease symptoms, and include SEIZURES, CVA/STROKE, OVERDOSE, ABDOMINAL PAINS, etc. It is known that STOKE is a common cause of SEIZURE (De Reuck, 2009) — in the aftermath of a stroke, the seizure is often experienced. Likewise, it is common that after OVERDOSES, people may feel ABDOMINAL PAINS. The detailed EMS titles in each cluster are listed in Table 1 of the supplementary material. Furthermore, from Figure 3c, we can see the cluster of townships tend to neighbor each other. This is reasonable, since one accident is more likely to cause subsequent accidents in adjacent geolocations. For example, a severe road accident may cause a traffic jam in a nearby town.

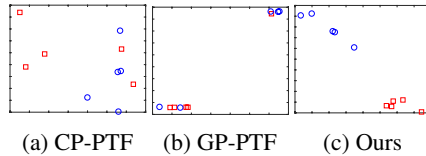

(a) CP-PTF    (b) GP-PTF    (c) Ours

Figure 2: The estimated latent factors in synthetic data.

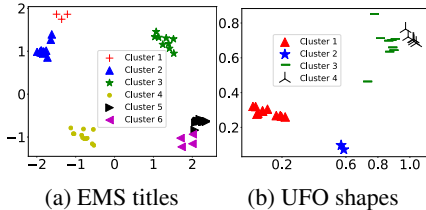

(a) EMS titles      (b) UFO shapes

![Townships map]

(c) Townships

Figure 3: Structures reflected from the latent factors learned by our model on *911* on *UFO*. In (c), the clusters of townships are shown in the actual map.

Second, we investigated the clusters of UFO shapes on *UFO* data (Fig. 3b). We found these clusters correspond to different appearance patterns. For example, Cluster 1 contain more three-dimensional looks, including *cone*, *cylinder*, *egg*, *pyramid*, etc, while Cluster 2 comprise thinner/flatter shapes, such as *disk* and *cigar*. Cluster 3 are {*fireball*, *flash*} and Cluster 4 are more about formation flying, such as *cross*, *delta* and *round*. The details are in Table 2 of the supplementary material. Generally, it reflects UFOs with similar looks are more likely to be sighted together/successively in a short time.

## 7 Conclusion

We proposed a nonparametric event-tensor decomposition model to capture the complex relationships and temporal dependencies in tensor data with time stamps. We developed a doubly stochastic variational EM algorithm for scalable inference. Our model has shown effectiveness on several real-world datasets. In the future, we will investigate more and larger scale applications.

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
