[Supplementary Material · Supplementary.pdf]

## Supplementary Materials

In this extra material, we supplement the details of the updates in our doubly stochastic variational EM algorithm in Section 1, and some experimental details in Section 2.

## 1 Updates of Doubly Stochastic Variational EM

We assume the variational posterior of the pseudo targets $\mathbf{g}$ is a multivariate Gaussian distribution, $q(\mathbf{g}) = \mathcal{N}(\mathbf{g}|\boldsymbol{\mu}, \boldsymbol{\Sigma})$. To ensure $\boldsymbol{\Sigma}$ is positive definite, we represent $\boldsymbol{\Sigma}$ with its Cholesky factorization, namely, $\boldsymbol{\Sigma} = \mathbf{L}\mathbf{L}^\top$ where $\mathbf{L}$ is a lower triangular matrix.

**E step**. Based on the stochastic objective in (14) of the main paper, we can derive that the optimal of variational posterior for each latent cause $q(z_j)(j \in N_k)$ is a multinomial distribution,

$$q(z_j = 0) \propto e^{\mathbb{E}_{q(\mathbf{g})}\mathbb{E}_{p(f_{\mathbf{i}_j}|\mathbf{g})}(f_{\mathbf{i}_j})}$$

$$q(z_j = n) \propto \beta k(\mathbf{x}_{\mathbf{i}_j}, \mathbf{x}_{\mathbf{i}_n})e^{\frac{1}{\tau}(s_j - s_n)}$$

where $n \in \bar{A}_{s_j}$, $p(f_{\mathbf{i}_j}|\mathbf{g})$ is a conditional Gaussian distribution,

$$p(f_{\mathbf{i}_j}|\mathbf{g}) = \mathcal{N}\big(f_{\mathbf{i}_j}|c(\mathbf{x}_{\mathbf{i}_j}, \mathbf{B})c(\mathbf{B}, \mathbf{B})^{-1}\mathbf{g}, c(\mathbf{x}_{\mathbf{i}_j}, \mathbf{x}_{\mathbf{i}_j}) - c(\mathbf{x}_{\mathbf{i}_j}, \mathbf{B})c(\mathbf{B}, \mathbf{B})^{-1}c(\mathbf{B}, \mathbf{x}_{\mathbf{i}_j})\big),$$

$\mathbf{x}_{\mathbf{i}_j}$ and $\mathbf{x}_{\mathbf{i}_n}$ are the latent factor concatenations for the entry $\mathbf{i}_j$ and $\mathbf{i}_k$, i.e.,

$$\mathbf{x}_{\mathbf{i}_j} = [\mathbf{U}^{(1)}(i_{j_1}, :), \ldots, \mathbf{U}^{(K)}(i_{j_K}, :)],$$

$$\mathbf{x}_{\mathbf{i}_n} = [\mathbf{U}^{(1)}(i_{n_1}, :), \ldots, \mathbf{U}^{(K)}(i_{n_K}, :)].$$

**M step**. In the M step, we calculate the gradient of the stochastic objective $\tilde{L}_{k,l}$ w.r.t to all the other parameters $\boldsymbol{\theta}$ in our model, including $\beta, \tau, \boldsymbol{\mu}, \mathbf{L}$, the covariance function parameters used in GPs and the latent factors $\mathcal{U} = \{\mathbf{U}^1, \ldots, \mathbf{U}^K\}$. We update these parameters by

$$\boldsymbol{\theta}^{t+1} = \boldsymbol{\theta}^t + \eta_t \frac{\partial \tilde{L}_{k,l}}{\boldsymbol{\theta}^t}$$

where $\boldsymbol{\theta}^t$ and $\boldsymbol{\theta}^{t+1}$ are the parameters $\boldsymbol{\theta}$ in step $t$ and $t+1$ respectively, and $\eta_t$ is the step-size. Note that when we use AdaDelta (Zeiler, 2012) framework, the step-size for each element of $\frac{\partial \tilde{L}_{k,l}}{\boldsymbol{\theta}^t}$ is adjusted separately. In doing so, the stochastic gradient descent often converges faster. The gradient w.r.t the base triggering function parameters $\beta$ and $\tau$ are given by

$$\frac{\partial \tilde{L}_{k,l}}{\partial \beta} = -\frac{N}{N_k}\frac{M}{M_l}\sum_{j \in N_k}\sum_{\mathbf{i} \in M_l} k(\mathbf{x}_{\mathbf{i}_j}, \mathbf{x}_{\mathbf{i}})\tau(1 - e^{\frac{1}{\tau}\Delta_j})$$

$$+ \frac{N}{N_k}\sum_{j \in N_k}\sum_{n \in \bar{A}_{s_j}} \mathbb{E}_{q(z_j)}\big(\mathbb{1}(z_j = n)\big)\beta^{-1},$$

$$\frac{\partial \tilde{L}_{k,l}}{\partial \tau} = -\frac{N}{N_k}\frac{M}{M_l}\sum_{j \in N_k}\sum_{\mathbf{i} \in M_l} \beta k(\mathbf{x}_{\mathbf{i}_j}, \mathbf{x}_{\mathbf{i}})(1 - e^{\frac{1}{\tau}\Delta_j} - \frac{1}{\tau}\Delta_j e^{\frac{1}{\tau}\Delta_j})$$

$$+ \frac{N}{N_k}\sum_{j \in N_k}\sum_{n \in \bar{A}_{s_j}} (s_j - s_n)\mathbb{E}_{q(z_j)}\big(\mathbb{1}(z_j = n)\big)\tau^{-2}$$

where $\Delta_j = \min\left(s_{\min(j+C_{\max},N)}, \min(s_n + \Delta_{\max}, T)\right) - s_j$. To guarantee that $\beta, \tau > 0$, we update $\beta$ and $\tau$ in the log domain. The gradient w.r.t. $\log(\beta)$ and $\log(\tau)$ are easily obtained by

$$\frac{\partial \tilde{L}_{k,l}}{\partial \log(\beta)} = \frac{\partial \tilde{L}_{k,l}}{\partial \beta} \cdot \frac{\partial \beta}{\partial \log(\beta)} = \frac{\partial \tilde{L}_{k,l}}{\partial \beta} \beta,$$

$$\frac{\partial \tilde{L}_{k,l}}{\partial \log(\tau)} = \frac{\partial \tilde{L}_{k,l}}{\partial \tau} \cdot \frac{\partial \tau}{\partial \log(\tau)} = \frac{\partial \tilde{L}_{k,l}}{\partial \tau} \tau.$$

The gradient w.r.t the parameters of $q(\mathbf{g})$ are given by

$$\frac{\partial \tilde{L}_{k,l}}{\partial \boldsymbol{\mu}} = -c(\mathbf{B}, \mathbf{B})^{-1}\boldsymbol{\mu} + \frac{N}{|N_k|} \sum_{j \in N_k} \left[ \mathbb{E}_{q(z_j)}\left(\mathbb{1}(z_j = 0)\right) c(\mathbf{B}, \mathbf{B})^{-1} c(\mathbf{B}, \mathbf{x}_{\mathbf{i}_j}) \right.$$

$$\left. - \frac{T}{n_{\mathbf{i}_j}} b_j \cdot c(\mathbf{B}, \mathbf{B})^{-1} c(\mathbf{B}, \mathbf{x}_{\mathbf{i}_j}) \right]$$

$$\frac{\partial \tilde{L}_{k,l}}{\partial \mathbf{L}} = \mathrm{tril}\Big[ -c(\mathbf{B}, \mathbf{B})^{-1}\mathbf{L} + \mathbf{L}^{-1^\top}$$

$$- \frac{N}{|N_k|} \sum_{j \in N_k} \frac{T}{n_{\mathbf{i}_j}} b_j \cdot c(\mathbf{B}, \mathbf{B})^{-1} c(\mathbf{B}, \mathbf{x}_{\mathbf{i}_j}) c(\mathbf{x}_{\mathbf{i}_j}, \mathbf{B}) c(\mathbf{B}, \mathbf{B})^{-1}\mathbf{L}\Big]$$

where $\mathrm{tril}[\cdot]$ takes the lower triangle part of the matrix, $n_{\mathbf{i}_j}$ is the number of interaction events in entry $\mathbf{i}_j$,

$$b_j = \exp\Big\{ \frac{1}{2}\big[ c(\mathbf{x}_{\mathbf{i}_j}, \mathbf{x}_{\mathbf{i}_j}) - c(\mathbf{x}_{\mathbf{i}_j}, \mathbf{B}) c(\mathbf{B}, \mathbf{B})^{-1} c(\mathbf{B}, \mathbf{x}_{\mathbf{i}_j})$$

$$+ c(\mathbf{x}_{\mathbf{i}_j}, \mathbf{B}) c(\mathbf{B}, \mathbf{B})^{-1} \mathbf{L}\mathbf{L}^\top c(\mathbf{B}, \mathbf{B})^{-1} c(\mathbf{B}, \mathbf{x}_{\mathbf{i}_j})\big] + c(\mathbf{x}_{\mathbf{i}_j}, \mathbf{B}) c(\mathbf{B}, \mathbf{B})^{-1}\boldsymbol{\mu}\Big\}.$$

Finally, the calculation of the gradient w.r.t the latent factors and the kernel parameters are similar to (Lawrence, 2004; Zhe et al., 2016). We refer the details to the papers.

## 2 Experimental Details

### 2.1 Simulation

To examined the capability of our model in discovering latent structures in data, we first simulated a small synthetic event-tensor, of size $10 \times 10 \times 10$. The latent factors in each mode were sampled from a Gaussian mixture model (GMM) with 2 components, where the centers are $\{(-1,-1), (1,1)\}$. Given the latent factors, we sampled Hawkes process events for each tensor entry. The base rate for each entry $\mathbf{i}$ was generated via a nonlinear function,

$$\lambda_{\mathbf{i}}^0 = 1/(1 + x^2 + x) + e^{-\cos(x)}$$

where

$$x = \|\mathbf{U}^{(1)}(i_1, :) - \mathbf{U}^{(2)}(i_2, :)\| + \|\mathbf{U}^{(1)}(i_1, :) - \mathbf{U}^{(3)}(i_3, :)\| + \|\mathbf{U}^{(2)}(i_2, :) - \mathbf{U}^{(3)}(i_3, :)\|.$$

The rate function is defined as

$$\lambda_{\mathbf{i}}(t) = \lambda_{\mathbf{i}}^0 + \sum_{s_n < t} 0.1 e^{-50\|\mathbf{x}_{\mathbf{i}} - \mathbf{x}_{\mathbf{i}_n}\|^2} \mathbb{1}(t - s_n < 0.01) e^{\frac{t - s_n}{10}}$$

where $\mathbf{x_i}$ and $\mathbf{x_{i_n}}$ are the vectors from concatenating the latent factors associated with $\mathbf{i}$ and $\mathbf{i}_n$, respectively; $\mathbf{i}_n$ is the entry index for $n$-th event. We used Hawkes Process Toolkit (Xu and Zha, 2017) to sample $1,000$ events in total. We then ran CP-PTF, GP-PTF and our approach for 50 epochs to estimate the latent factors. All the methods started with the same random initialization of the latent factors.

## 2.2 Running Time

We implemented our algorithm, GP-PTF and CP-PTF with Matlab. We ran all the methods on a desktop machine with Intel i7 2.90GHz processors. The average running time per epoch of our approach are $1.75$, $6.3$ and $13.67$ minutes for *911*, *UFO* and *Article* datasets, respectively. The average per-epoch running time for GP-PTF and CP-PTF are much faster, which are less than 1 minute for all the datasets. This is reasonable, because GP-PTF and CP-PTF only need to process each observed entry, where the interaction events are simply aggregated into a single count. By contrast, our model needs to process every single event inside each entry (see (6) and the equation above in the main paper), and hence the number of data points are much larger. In addition, to capture the dependency among the interactions, our model further has to consider the proceeding events in the triggering function and HP likelihoods (see (4) in the main paper). Nevertheless, the complexity of our doubly stochastic variational EM algorithm is only proportional to the production of the event and entry batch sizes (see Sec. 4.3 of the main paper), and hence are still scalable to handle both many observed entries and events.

## 2.3 Clusters of Latent Factors

We used the k-means algorithm to cluster our estimated latent factors for *911* and *UFO* datasets. For visualization, we set the number of latent factors to 2. We chose the latent factors that gave the best prediction performance among the 100 epochs. We used BIC to identify the appropriate cluster number from 2 to 15. Finally, we obtained 10 groups of townships and 6 groups of EMS titles for *911*, and 4 groups of UFO shapes for *UFO*. The clusters for EMS titles and UFO shapes are listed in Table 1 and 2, respectively.

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

Table 1: The clusters of the latent factors for EMS titles

| | Members |
| --- | --- |
| Cluster 1 | SHOOTING, AMPUTATION, S/B AT HELICOPTER LANDING |
| Cluster 2 | CVA/STROKE, SEIZURES, OVERDOSE, ABDOMINAL PAINS, BACK PAINS/INJURY, DIABETIC EMERGENCY, RESPIRATORY EMERGENCY, SYNCOPAL EPISODE , LACERATIONS, FEVER, ALLERGIC REACTION, RESCUE - GENERAL, FIRE SPECIAL SERVICE, DEHYDRATION, POISONING, CARDIAC ARREST |
| Cluster 3 | VEHICLE FIRE, FIRE INVESTIGATION, MEDICAL ALERT ALARM, FIRE ALARM, TRANSFERRED CALL, RESCUE - WATER, DEBRIS/FLUIDS ON HIGHWAY, DISABLED VEHICLE, SUSPICIOUS |
| Cluster 4 | EMS SPECIAL SERVICE, CARBON MONOXIDE DETECTOR, BUILDING FIRE, APPLIANCE FIRE, RESCUE - TECHNICAL, ELECTROCUTION, STABBING, HAZARDOUS MATERIALS INCIDENT, RESCUE - ELEVATOR, ACTIVE SHOOTER, BOMB DEVICE FOUND, UNKNOWN TYPE FIRE |
| Cluster 5 | CARDIAC EMERGENCY, DIZZINESS, HEAD INJURY, NAUSEA/VOMITING ALTERED MENTAL STATUS, SUBJECT IN PAIN, HEMORRHAGING FALL VICTIM, MATERNITY, UNCONSCIOUS SUBJECT, CHOKING FRACTURE, BURN VICTIM, EYE INJURY, HEAT EXHAUSTION |
| Cluster 6 | GAS-ODOR/LEAK, TRAIN CRASH, PLANE CRASH, WOODS/FIELD FIRE |

Table 2: The clusters of the latent factors for UFO shapes

| | Members |
| --- | --- |
| Cluster 1 | cone, cylinder, diamond, egg, chevron, circle, crescent, pyramid, sphere, teardrop, triangle |
| Cluster 2 | fireball, flash |
| Cluster 3 | cigar, disk, flare, hexagon, light oval, rectangle, unknown, changed |
| Cluster 4 | formation, cross, delta, changing, round, other |