[Reviews · NeurIPS 2018]

Reviewer 1



The work proposed a nonparametric Bayesian model for event tensor decomposition. Existing tensor works lack of a way to integrate the complete temporal information in the factorization. This work formulates the so called “event-tensor” to preserver all the time stamps, where each entry consists of a sequence of events rather than a value. To decompose the event-tensor, the work hybridizes Gaussian processes and Hawkes processes to model the entries as mutually excited Hawkes processes, and base rates Gaussian processes on the latent factors. The authors exploit the Poisson process super-position theorem and variational sparse GP framework to derive a decomposable variational lower bound and develop a doubly stochastic algorithm for scalable inference. The method is examined on both simulation and real datasets, exhibits superior predictive performance to commonly used factorization methods, and finds interpretable discoveries. This is a well-written paper, and I enjoy reading it. The work is very interesting --- temporal information are common in practical applications, and how to effectively capture rich dependency structures from these temporal information, and use them in tensor decomposition is important. This work is the first one to address this issue, and the proposed model is novel and interesting. I particularly like the idea that incorporates the latent factors into the design of the triggering kernel so that those excitation effects can be encoded into the latent factors as well. The inference algorithm is also scalable. The derived variational lower bound has an interesting structure that can be decomposed over both tensor entries and interaction events; so the authors can develop a doubly stochastic estimation algorithm. The experimental results show interpretable discoveries on 911 and ufo data. Overall, since the proposed data formulation exists in a wide variety of applications, this work seems to have a broad spectrum of usage. Here are a few minor questions: 1. For a decomposition model, I would like to know how to reconstruct missing entries? In this work, it would be missing event sequence 2. It seems like this work can be extended to general embedding learning approaches for high-order interactions/relations. What do you consider?

Reviewer 2



Post rebuttal: thank you for your answers! I would encourage authors to consider experiments with matrix completion with time stamps in the camera ready version and compare to baselines such as Factorization Machines or Neural Networks where time information is ignored. In my opinion this could help to draw the attention of researchers outside of the Bayesian community and increase the significance of the work. This paper considers the problem of tensor factorization across time. Proposed model combines Hawkes process to accommodate temporal dependencies, with base rate dependent on a function of latent factors equipped with Gaussian Process prior. Gaussian Process has been previously considered in the tensor factorization domain and showed some advantages over more classical multilinear functions. However the addition of Hawkes process while maintaining advantages of GP appear novel to me and more elegant than previous discrete time approaches. Authors take Variational Inference pass, but deriving ELBO is not trivial for the proposed model and as a workaround authors decompose Hawkes process into multiple Poisson processes and introduce additional categorical ("cause") latent variable to arrive at manageable ELBO. I think this is a good paper addressing a fairly sophisticated problem of stochastic tensor decomposition with a new model and some interesting inference decisions. Combination of Hawkes process and GP was new to me and interesting to see. Experimental baselines appear strong and experiments demonstrate the advantage of modeling time component and using GP. Overall paper is written well. On the downside, method is quite slow, but nonetheless experimental results are shown on several relatively big datasets. The other downside is the practical usefulness of the method. Clustering analysis does not appear particularly meaningful (looking at the Tables in the supplement, I can't really relate to the interpretation suggested in the paper), although perhaps the data is to blame (method seems to identify some structure in the toy simulation experiment). Have you tried matrix completion? For example, 100k MovieLens data (for predicting user-movie interactions instead of actual ratings)? I believe timestamp is often discarded, but maybe your model can perform well by taking it into account. It would be interesting to know if the method can compete with black box prediction approaches (i.e. neural networks or Factorization Machines). It is ok if not, since there are other merits, but it could make the story more complete and show limitations of the approach (overall, authors did not really discuss what are the limitations besides run-time). Experiments show that GP + Hawkes process are better than GP or multilinear factorization with discrete time. What about using discrete time approach and GP instead of multilinear factorization. Is there any prior work that uses GP (for tensor decomposition) in temporal setting? Typos in lines 55, 377.

Reviewer 3



This paper develops a Bayesian nonparameteric tensor factorization model with the goal of fully incorporating the temporal information, i.e. without discretization, and also capturing the nonlinear relationships in the data. This is achieved by defining "event-tensor", whose entries comprise sequences of interaction events rather than numerical values representing the interaction counts. This tensor-events are then modeled using the Hawkes process, which has the capability of capturing the triggering effects. Major comments: 1. How the triggering effects between two different users can be captured by this model? Can you provide an argument based on the data and model that shows how the similar users end up having close latent factors, while the interaction between users are unknown? 2. Isn't it better to define A_t in (4) based on time proximity rather than event counts, so that related events are not discarded because of presence of other non-triggering events in that time interval? 3. In your experiments for figure 1, wasn't it a better idea to set C_max to different values, rather than \Delta_max? Isn't this the reason that the results aren't that different? Minor comments: 1. In line 63, shouldn't the matrix U be t by s, to be consistent with the matrix product notations?